# What research evidence exists about physical activity in parents? A systematic scoping review

Rachel F Simpson [ORCID],[1] Kathryn R Hesketh,[1,2] Kate Ellis,[1] Esther MF van Sluijs[1]

[1]MRC Epidemiology Unit, Cambridge, UK
[2]UCL Great Ormond Street Institute of Child Health, UCL, London, UK

**Correspondence to**
Dr Rachel F Simpson;
rachel.simpson@mrc-epid.cam.ac.uk

## ABSTRACT

**Objectives** Despite the known benefits of physical activity (PA) to physical and mental health, many people fail to achieve recommended PA levels. Parents are less active than non-parent contemporaries and constitute a large potential intervention population. However, little is known about the breadth and scope of parental PA research. This scoping review therefore aimed to provide an overview of the current evidence base on parental PA.

**Methods** Four databases (MEDLINE, Embase, PsycINFO and Scopus) were systematically searched to identify peer-reviewed articles focusing on parental PA from 2005 onwards, including interventional, observational or qualitative study designs. Title and abstract screening was followed by duplicate full-text screening. Data extracted for all articles (100% checked by a second reviewer) included study design, proportion of fathers and ages of children. For interventional/observational studies, PA assessment method and factors examined or targeted based on the socio-ecological model were extracted, and questions addressed in qualitative studies.

**Results** Of 14 913 unique records retrieved, 213 articles were included; 27 articles reported on more than one study design; 173 articles reported on quantitative (81 cross-sectional, 26 longitudinal and 76 interventional) and 58 on qualitative data. Most articles originated from North America (62%), and 53% included only mothers, while 2% included only fathers. Articles most frequently represented parents of infants (56% of articles), toddlers (43%), preschoolers (50%) and primary-school aged children (49%). Most quantitative articles only reported self-reported PA (70%). Observational articles focused on individual correlates/determinants (88%). Likewise, most interventions (88% of articles) targeted individual factors. Most qualitative articles explored PA barriers and facilitators (57%).

**Conclusions** A range of quantitative and qualitative research has been conducted on parental PA. This review highlights opportunities for evidence synthesis to inform intervention development (such as barriers and facilitators of parental PA) and identifies gaps in the literature, for example, around paternal PA.

**Review registration** osf.io/qt9up.

## Strengths and limitations of this study

► This scoping review encompasses a wide range of study designs and provides detailed information on the research conducted to date.
► It was conducted rigorously and systematically following a predefined protocol and guidance for scoping reviews.
► Despite this, some articles may have been missed as is the case for all reviews.
► Articles relating to parents or children in a clinical population were not included.
► Definitions of parents vary across the literature, but most definitions were similar to each other.

various cancers and depression, improves bone health and is beneficial for weight maintenance.[2] There is also evidence that physical activity can mitigate the negative effects of sedentary behaviour, which is unavoidable during daily life for many people.[3] Despite this, 28% of adults worldwide are insufficiently physically active, defined as participating in less than 150 min of moderate or vigorous physical activity per week.[4]

Considering the importance of physical activity, it is vital to find ways to increase physical activity levels in populations at higher risk of insufficient activity levels. One such population is parents, herein defined as mothers or fathers of children, whether they are biological parents, adoptive, foster or step-parents. Review-level evidence has shown that parenthood is negatively associated with adult activity levels,[5] and specifically that parents are less active than non-parents,[6] mothers are less active than non-mothers[7] and fathers engage in less moderate-to-vigorous physical activity than non-fathers.[8]

As well as general population health benefits, physical activity could have parent-specific and family-specific benefits. Parent-specific benefits include an improved ability to cope with both the physical and emotional requirements of daily life as a parent,[9] and

## INTRODUCTION

Regular engagement in physical activity has many benefits for physical and mental health.[1] It reduces the risk of cardiovascular disease,

closer parent–child relationships through shared interests and co-participation (used interchangeably with co-activity in the literature).[10] Parental physical activity has also been linked to that of their children, especially in studies using device-based assessment.[11–13] Increasing physical activity levels of parents could therefore lead to increases in that of their children through various mechanisms including modelling,[11] co-participation,[14] appreciation of the importance of physical activity from a young age and creation of a family environment conducive to all family members being active.[15]

The scope of parental physical activity research is very broad and could be quantitative or qualitative, focused on mothers and/or fathers, on parents of children of a specific age group or on a particular subpopulation of parents, such as working parents or single parents. Most recent systematic reviews conducted in relation to parental physical activity have focused on narrow sets of studies. These include interventions to improve child and parental physical activity[16 17] and physical activity in postpartum women,[18] and the qualitative experiences of postpartum women.[19] Three broader quantitative systematic reviews have been conducted in relation to parental[6] or maternal[7] physical activity (both in 2008), or paternal physical activity (in 2016).[8] However, the systematic review on paternal physical activity only focused on studies comparing physical activity levels of fathers and non-fathers, rather than also examining differences in physical activity levels between fathers. To our knowledge, no broader qualitative systematic review has been conducted.

Several limitations in studies relating to parental physical activity were identified by Bellows-Riecken and Rhodes in their systematic review in 2008, the only one of the three broader systematic reviews including articles relating to both maternal and paternal physical activity. Only 8 of the 31 articles included fathers, only 1 study used a device-based assessment (pedometer) of physical activity, and possible environmental or wider-societal correlates and determinants according to the socio-ecological model (SEM) were under investigated.[6] The SEM, as used in the Bellows-Riecken and Rhodes *et al* 2008 review,[6] provides a useful framework with which to examine factors that influence the physical activity of parents, as having children could influence all levels of the model. It consists of individual (biological/demographic, behavioural and psychological), interpersonal (social support and influence of family, friends and social networks), environmental (social, built and natural) and wider societal factors (social and cultural norms and public policy) which can overlap.[20–22] The present review aims to identify to what extent the limitations identified by Bellows-Riecken and Rhodes have been addressed, but also explores the qualitative literature and how many studies have been carried out with parents of children of various ages, since parenting children of different age groups brings its own unique experiences and challenges.

An overview of both the current quantitative and qualitative evidence available in relation to parental physical activity would be advantageous to inform future research. A scoping review is a suitable way of achieving this.[23] Qualitative and quantitative research are complementary to each other. They answer different questions about physical activity, for example, quantitative research can tell us what factors are associated with parental physical activity, while qualitative research can tell us why this is the case. A scoping review also provides a means to determine whether there is scope and a need for a systematic review of a particular type or in a certain area of literature, and to identify gaps in the existing research base which could be filled by future primary research.[23]

The overall aim of this scoping review is therefore to determine the extent and nature of the literature exploring physical activity levels in parents. This includes study designs used, ages of children, representation of fathers, assessment method of physical activity, factors in quantitative studies which could influence parental physical activity, according to the SEM and questions addressed in the qualitative literature.

## METHODS

The framework for this scoping review was based on that proposed by Arksey and O'Malley[23] and elaborated on by Levac *et al*[24] and the Joanna Briggs Institute.[25] Thus, the review contained five stages:

1. Identifying the research question.
2. Identifying relevant studies.
3. Selecting studies.
4. Charting the data.
5. Collating, summarising and reporting the results.

Arksey and O'Malley also include a stage 6 which is a consultation exercise with key stakeholders,[23] but this is optional and was not conducted.

The Preferred Reporting Items for Systematic Reviews and Meta-Analyses extension for Scoping Reviews (PRISMA-ScR) was used as a guideline for writing this paper (see online supplemental file 1 for the completed checklist).[26] The review protocol (see online supplemental file 2) was prospectively registered with Open Science Framework.

### Stage 1—identifying the research question

Based on an exploratory review of the literature, the overarching research question identified was 'What is the extent and nature of the literature exploring physical activity in parents?'. Box 1 shows the sub-questions identified a priori. Observational, interventional and qualitative articles refer to articles relating to observational, interventional and qualitative study designs, respectively.

### Stage 2—identifying relevant studies

A literature search was conducted in four databases (MEDLINE, Embase, Scopus and PsycINFO) to provide access to a comprehensive range of interdisciplinary articles. The search was conducted from 2005 up to and

> Box 1   Sub-questions identified for the parental physical activity scoping review

1. How many observational, interventional and qualitative articles have been conducted on this topic?
2. What are the characteristics of the populations that have been investigated (eg, country, ethnicity, employment status, marital status)?
3. How many quantitative articles include device-assessed physical activity and how many self-reported measures?
4. How many quantitative articles have made comparisons between the physical activity levels of parents and non-parents; comparisons in physical activity levels among parents according to various factors; or both?
5. For each of the three types of article, how many articles have investigated physical activity of only fathers; only mothers; or both fathers and mothers?
6. Parents of children of what age groups have been investigated for each article type?
7. What factors according to the socio-ecological model have been investigated in quantitative articles regarding a possible association with parental physical activity?
8. What questions have been addressed by qualitative articles?

including 11 April 2020. The start date was selected to provide information most relevant to the present day, including factors associated with physical activity in parents within current society (since social norms change in relation to roles of mothers and fathers and the importance of physical activity). Peer-reviewed journal articles relating to observational, interventional, qualitative or mixed-methods studies in humans were considered. The search was limited to articles published in English.

The search strategy was developed in consultation with an academic librarian. A pilot search was conducted and reviewed to assess sensitivity and specificity of the search, with terms adjusted accordingly. A mixture of in-text words, keywords and Medical Subject Headings terms for the population (ie, parents) and the behaviour (ie, physical activity) were combined. Terms relating to the postpartum period were included in order to ensure that studies examining physical activity of new mothers were included (see online supplemental file 3 for database search terms). Finally, relevant systematic reviews were screened for appropriate studies.[6–8]

### Inclusion and exclusion criteria
A summary of the inclusion and exclusion criteria is outlined in table 1. Online supplemental file 4 provides a more detailed description of the criteria applied.

### Stage 3—study selection
Study results were exported to Covidence for de-duplication. A calibration exercise with 1000 titles and abstracts screened in duplicate showed satisfactory agreement. Therefore, single title and abstract screening was

| **Table 1** | Article inclusion and exclusion criteria for the scoping review of parental physical activity | |
|---|---|---|
| | **Included** | **Excluded** |
| Population | ► Articles including at least one parent, who must be generally healthy, of children, where children are defined as being people aged 0–18 years old. | ► Articles in which either the parents or children are part of a clinical population.<br>► Articles where it is likely that parents only have children older than 18 years if the ages of the children are not specified in the inclusion criteria. |
| Study design | ► Quantitative (observational, including longitudinal and cross-sectional, or interventional), qualitative or mixed methods. | ► Any other study design. |
| Intervention | ► In the case of interventional articles, include any type of intervention, as long as one of the main outcomes examined in the paper is parental physical activity. | ► Interventional articles in which parental physical activity is not a main outcome in the paper. |
| Comparisons | ► For quantitative articles, include comparisons between the physical activity levels of parents and non-parents, or comparisons in physical activity levels among parents according to various factors. | ► Comparisons between parents and non-parents in populations of pregnant women. |
| Focus/outcomes | ► For quantitative articles, assess physical activity levels of at least one parent, either using device-assessment or self-assessment methods. This includes mention of duration or frequency of physical activity.<br>► For qualitative articles, there is a wider remit for inclusion, with studies eligible if they investigate parents' feelings towards or experiences of their own physical activity. The purpose of the study had to be related to physical activity or a significant proportion of the paper had to focus on physical activity.<br>► In all articles, a main focus must be on the physical activity of parents. | ► Postpartum articles where the only measures of physical activity after pregnancy are taken when the participants could potentially be 12 weeks or less after birth based on the recruitment criteria (or where postpartum qualitative studies are conducted when any women could be 12 weeks or less postpartum).<br>► Articles investigating the association between parent/child physical activity unless there is also a focus on parental physical activity.<br>► Articles in which parental physical activity is explicitly a secondary outcome. These articles were not deemed to have a main focus on parental physical activity. |
| Publication type | ► Full peer-reviewed articles in academic journals. | ► All other types of publication. |
| Publication year | ► 2005 onwards. | ► Before 2005. |

conducted for the remaining articles. A 10% random sample of excluded studies was checked for consistency. As a high volume of articles was identified initially, a second duplicate title and abstract screening of this set was conducted with greater specification of exclusion and inclusion criteria to minimise articles requiring full-text screening. The full-texts of the remaining articles were retrieved and screened in duplicate. No additional articles were identified from the aforementioned systematic reviews.[6–8] Discrepancies at the full-text stage were resolved through discussion or by arbitration by a third reviewer if needed. As the purpose of a scoping review is not to evaluate the quality of relevant research conducted, an optional assessment of study bias was not conducted here.[23]

### Stage 4—charting the data

Data were extracted into Microsoft Excel spreadsheets. The data extraction template was pilot tested in duplicate for 10 articles and finalised through iterative adjustment. Data from remaining articles were extracted by one author and then checked in full by a second. Only factors targeted in interventions according to the SEM (with criteria for classifications adapted from Golden and Earp[27]—see online supplemental file 5) were extracted in duplicate. Disagreements were resolved by discussion or by arbitration by a third author. Extracted data for all articles included author, year of publication, study name or description of study, study design, study population description, number and percentage of fathers, sample size and range of ages of children of parents represented. For all quantitative articles, additional extracted data included whether comparisons were made between physical activity levels of parents and non-parents or between physical activity levels among parents by various factors, whether physical activity was self-reported or device-assessed and what tool or device was used for assessment. For observational articles, potential correlates or determinants of physical activity examined were extracted. For interventional articles, whether each article related to a main trial or a pilot/feasibility study, target population of the intervention (parents only, parents and children or children), theory on which the intervention was based, description of the intervention, and factors targeted according to the SEM were extracted. For qualitative articles, methodology used, and questions addressed relating to parental physical activity were extracted.

### Stage 5—collating, summarising and reporting results

Narrative methods, tabulation and graphs were used to summarise results relevant to the research questions by study design. Articles could be counted in multiple categories, for example, those articles which included both observational and qualitative analyses are included under both 'observational' and 'qualitative' articles, and articles representing parents of children ≤12 months old, 1–2 years-old and 3–4 years-old are counted in each of these categories.

### Changes made to the original protocol

Some minor changes were made to the published protocol in order to ensure that the purpose of this scoping review was best met. The protocol was adapted to focus on the number and characteristics of individual articles rather than studies due to the large number of records retrieved and greater suitability of this approach for a scoping review. A further research question was also added to examine the correlates or determinants investigated in observational articles and the factors targeted in interventional articles to better elucidate the nature and extent of research conducted on parental physical activity (box 1).

Given the volume of studies already identified through database searching, reference lists and citations of included articles were not checked. The description of inclusion and exclusion criteria was revised to clarify and provide greater specificity than those criteria initially included in the protocol and to ensure that included studies were addressing the purpose of the scoping review.

### Patient and public involvement

Patients or the public were not involved in the design, or conduct, or reporting, or dissemination plans of our research.

### RESULTS

A flow-chart of articles was created in accordance with PRISMA-ScR[26] (see figure 1). Of the 213 articles included (figure 1), 27 reported on more than one study design.

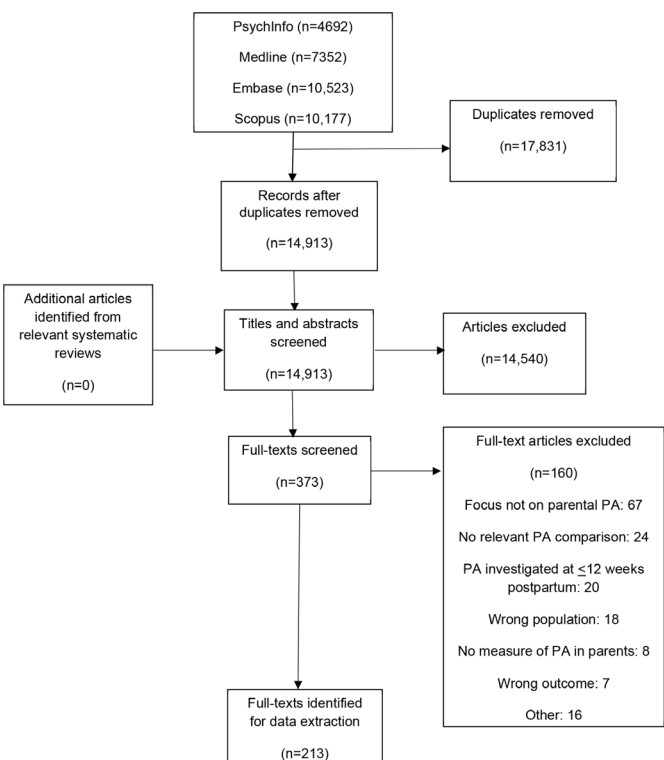

**Figure 1** A flow diagram of the screening and selection process in the scoping review. PA, physical activity.

**Table 2** Study characteristics and populations included in the parental physical activity scoping review by type of article*

| | Observational (n=99) | Interventional (n=76) | Qualitative (n=58) | Overall (n=213) |
|---|---|---|---|---|
| **Date published** | | | | |
| 2005–2010 | 18 (18%) | 19 (25%) | 17 (29%) | 51 (24%) |
| 2011–2015 | 41 (41%) | 27 (36%) | 16 (28%) | 77 (36%) |
| 2016–2020 | 40 (40%) | 30 (39%) | 25 (43%) | 85 (40%) |
| **Sample size** | 1–100: n=17 | 1–100: n=49 | 1–15: n=15 | |
| | 101–500: n=45 | 101–500: n=20 | 16+: n=41 | |
| | 501–1000: n=13 | 501–1000: n=4 | NR: n=2 | |
| | 1001+: n=24 | 1001+: n=3 | | |
| **Continent** | | | | |
| Asia | 4 (4%) | 5 (7%) | | 9 (4%) |
| Australasia | 13 (13%) | 14 (18%) | 19 (33%) | 43 (20%) |
| Europe | 12 (12%) | 10 (13%) | 5 (9%) | 27 (13%) |
| North America | 69 (70%) | 47 (62%) | 34 (59%) | 133 (62%) |
| Multiple continents | 1 (1%) | | | 1 (<1%) |
| **Target population characteristics** | | | | |
| Low-income parents | 8 (8%) | 17 (22%) | 10 (17%) | 32 (15%) |
| Single parents | 2 (2%) | 1 (1%) | 2 (3%) | 4 (2%) |
| Ethnic minorities | 10 (10%) | 11 (14%) | 6 (10%) | 24 (11%) |
| Parents with overweight/obesity | 4 (4%) | 8 (11%) | 1 (2%) | 13 (6%) |
| Working parents | 4 (4%) | 4 (5%) | 5 (9%) | 10 (5%) |
| Rural | 2 (2%) | 2 (3%) | 3 (5%) | 7 (3%) |
| Parents of children with overweight/obesity | 3 (3%) | 5 (7%) | 1 (2%) | 8 (4%) |

*Numbers are not exclusive—articles can be counted multiple times. Articles containing more than one type of analysis, for example, observational and interventional, are included under both article types.
NR, not reported.

One hundred and seventy-three (81%) articles reported on quantitative study designs (81 cross-sectional, 26 longitudinal and 76 interventional) and 58 (27%) articles reported on qualitative data. Online supplemental files 6-11 show the reference list of included articles and the extraction tables.

### Descriptive characteristics and methodological information of included articles

Across all articles, publication was spread across 2005–2020, and sample sizes varied by article type (table 2). Most studies were conducted in North America (62% of articles), with the remainder originating from Australasia, Asia and Europe (table 2). No studies had been conducted in South America or Africa. Physical activity of low-income parents was explored in 8% of observational articles, 22% of interventional articles and 17% of qualitative articles. Ethnic minorities were the target population in 11% of articles, and parents with overweight or obesity in 6% of articles (table 2). One observational article investigated physical activity of homosexual couples,[28] while another two observational[29 30] and one interventional article[31] focused on adolescent parents.

Most quantitative articles only contained self-reported physical activity without any device assessment (73% of observational and 66% of interventional). When device-assessment was used, accelerometry was most frequent (78% of observational and 62% of interventional articles), followed by pedometer-assessment (19% of observational and 38% of interventional articles) and other devices (4% of all articles using device-assessment). Most (83%) observational articles investigated correlates and determinants of physical activity among parents, 2% investigated differences in physical activity levels between parents and non-parents and 15% investigated both. The majority of interventional articles described full trials (71%), rather than pilot or feasibility studies (29%), and involved parents alone (59%), rather than children and parents (39%) or children alone (1%), and mentioned a theory on which the intervention was based (66%).

The most common methodologies used in qualitative articles were individual interviews (45%) and focus groups or group interviews (47%). Other methodologies included open-ended questions on a survey (16%), event history calendars (2%), family unit interviews (3%),

**Table 3** Representation of mothers and fathers and age groups of their children in a parental physical activity scoping review by type of article*

| | Observational n=99 | Interventional n=76 | Qualitative n=58 | Overall n=213 |
|---|---|---|---|---|
| Mothers only, fathers only or both represented | | | | |
| Mothers only | 41 (41%) | 48 (63%) | 35 (60%) | 113 (53%) |
| Fathers only | 1 (1%) | 2 (3%) | 1 (2%) | 4 (2%) |
| Both fathers and mothers | 57 (58%) | 26 (34%) | 22 (38%) | 96 (45%) |
| Proportion of fathers when both mothers and fathers represented (median %, IQR) | 42% (24%–48%) | 15% (7%–32%) | 33% (23%–48%) | 33% (15%–48%) |
| Age groups covered by age ranges of children | | | | |
| Infants (≤12 months old) | 55 (56%) | 40 (53%) | 35 (60%) | 119 (56%) |
| Toddlers (1–2 years-old) | 42 (42%) | 23 (30%) | 34 (59%) | 91 (43%) |
| Preschoolers (3–5 years-old) | 52 (53%) | 27 (36%) | 37 (64%) | 106 (50%) |
| Primary-school aged (5–12 years-old) | 55 (56%) | 32 (42%) | 30 (52%) | 104 (49%) |
| Young adolescents (12–15 years-old) | 37 (37%) | 15 (20%) | 16 (28%) | 60 (28%) |
| Older adolescents (16–18 years-old) | 26 (26%) | 8 (11%) | 11 (19%) | 40 (19%) |

*Numbers are not exclusive—articles can be counted multiple times. Age groups of children are those covered by the age range of the children of parents in the articles, for example, an age range of 0–18 years-old would be included in all of the age groups, and 3–8 years-old would be included in preschoolers and primary-school-aged.
NR, not reported.

stories and images (2%) and verbal feedback (2%). Seven articles included multiple data collection methodologies.

### Representation of mothers, fathers and children's ages

Table 3 shows that most observational articles included both fathers and mothers (58%), while the majority of interventional (63%) and qualitative (60%) articles included only mothers. Less than 4% of all articles included fathers only. When articles included both mothers and fathers, fathers were usually under-represented: in the 90 articles including both mothers and fathers and reporting the percentage of fathers, the median was 33% (IQR 15%–48%), with the lowest representation of fathers in interventional articles (median 15%, IQR 7%–32%). Articles in which eligible parents had children ≤12 years old were more common than those in which parents of older children were represented. Of the 86 articles relating to parents of a specific age group of children, infants (52%) and primary-school aged (36%) were the most common age groups, with none relating specifically to parents of toddlers or older adolescents (see online supplemental file 12).

### Socio-ecological factors examined or targeted in quantitative articles

Figure 2 presents the number of observational articles in which individual (biological/demographic; behavioural; psychosocial), interpersonal (child-factors; partner-factors; family-factors; friend/colleague factors) and environmental (social; built; natural) correlates or determinants were examined in relation to parental physical activity. No wider societal factors were identified in observational articles. Online supplemental file 8 presents the

specific factors examined in individual observational articles.

Most articles (88%) examined individual factors, followed by interpersonal (eg, gender of child and partner support) (29%), and then environmental (eg, aesthetics and road safety) (26%). The most commonly investigated individual factors were biological or demographic (eg, age of parent, parental body mass index (BMI), household income) (included in 66% of all observational articles), followed by psychosocial factors (eg, self-efficacy and enjoyment of physical activity) (included in 39% of observational articles). The least investigated individual factors were behavioural (eg, past parental physical activity behaviour and breastfeeding status) (included in 17% of articles).

Figure 3 represents the number of interventional articles in which the various levels of the SEM model were

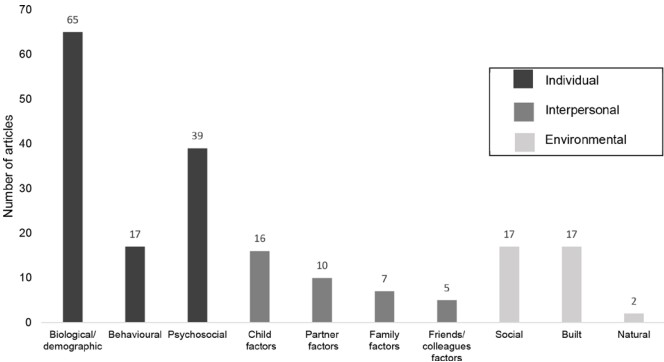

**Figure 2** Number of observational articles investigating different types of socio-ecological correlates or determinants (total n=99).

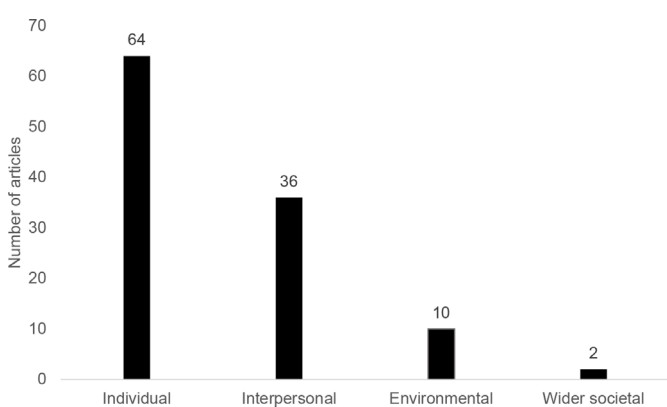

**Figure 3** Number of interventional articles with interventions targeting different levels of the socio-ecological model (n=73). Three interventional articles were not included as they were about mediators rather than the effect of the intervention itself.

targeted. Most interventions (88% of 73 articles) targeted individual factors (eg, self-efficacy and knowledge of physical activity), followed by interpersonal (49%) (eg, social support provision), environmental (14%) (eg, changes to the physical or social environment, such as modifications to the home or work environment) and

then wider societal (3%) (eg, public policy) (see online supplemental file 9 for details by study).

### Questions addressed in qualitative articles

Table 4 shows the range of questions explored in qualitative articles along with examples. The most commonly explored question area was barriers, facilitators and motivators to physical activity (n=33, 57%). Details of specific question areas covered by individual articles are included in online supplemental file 10.

## DISCUSSION
### Main findings

This systematic scoping review shows that a wide range of research has been conducted in relation to parental physical activity since 2005. Few articles have focused specifically on paternal physical activity and even where fathers are included, they tend to be under-represented as a proportion of parents in the overall sample size. Research tends to focus more on parents of children ≤12 years old, compared with those with older children, and most parental physical activity research has been conducted in North America, with no evidence from South America or Africa. Some articles have focused

**Table 4** Question areas explored in qualitative articles in the parental physical activity scoping review*

| Question area | Examples of questions or topics covered | Number of articles out of 58 (%) |
| --- | --- | --- |
| Barriers, facilitators and motivators (including strategies or changes to increase physical activity) | 1. What, if anything, might keep you from doing your exercise over the next week or few weeks?<br>2. What do you see as the advantages of your doing regular moderate physical activity? | 33 (57%) |
| Identity and perceptions and meaning of physical activity | 1. Do parents consider themselves to be active since having children?<br>2. Mothers' beliefs about the importance of a healthy diet and exercise. | 25 (43%) |
| Intervention or programme-related questions | 1. What to include in a parental physical activity intervention.<br>2. Acceptability of the intervention. | 22 (38%) |
| Physical activity patterns and experiences of physical activity as a parent | 1. When do you walk?<br>2. Expectations and experiences about physical activity. | 15 (26%) |
| Effect of the environment | 1. What types of opportunities exist in their community for being physically active?<br>2. Social and environmental factors influencing family co-participation in physical activity. | 8 (14%) |
| Co-participation related | 1. Reciprocal familial influences on co-participation in physical activity.<br>2. Parental beliefs about co-activity. | 8 (14%) |
| Influence of others on physical activity | 1. Parental thoughts on getting assistance from others to do regular physical activity.<br>2. Who are the individuals or groups of people that would approve or want you to do regular moderate physical activity? | 7 (12%) |
| Changes to physical activity since having children | 1. Do they feel that your physical activity level has changed since having children?<br>2. Changes in the types of physical activity since having children. | 6 (10%) |
| Understanding of physical activity | 1. What counts as valid physical activity and walking?<br>2. How do mothers categorise physical activity? | 5 (9%) |
| Other (sources of information, lifetime physical activity changes, etc) | 1. The mother's sources of health information and whether she values these sources.<br>2. Changes in physical activity since childhood. | 3 (5%) |

*Numbers are not exclusive—articles can be counted multiple times.

on specific populations, in particular those of low socio-economic status and ethnic minorities, but most articles relate to the general population. Although both quantitative and qualitative research has been carried out, more than twice the number of quantitative articles have been published.

The quantitative literature is dominated by cross-sectional or interventional articles and self-reported physical activity. Individual-level factors have been most commonly explored in observational studies, especially biological or demographic factors. Interventions have also tended to target individual-level factors, in particular psychosocial factors. The majority of the qualitative literature has focused on barriers, facilitators and motivators for parental physical activity, although a wide range of questions have been addressed.

## Findings in relation to relevant literature

Bellows-Riecken and Rhodes[6] stated that greater representation of fathers in parental physical activity research was required, but this scoping review highlights that fathers remain under-represented in the literature, especially in interventional articles. Given that physical activity levels of both fathers and mothers have been found to be lower than non-parents,[7 8] it is necessary to find ways to increase physical activity levels of parents of both genders. Evidence suggests that mothers and fathers share similar barriers, facilitators and experiences of physical activity as a parent.[32] However, differences by gender also exist, for example, around attitudes about the importance of types of social support for physical activity,[33] and the use of the lunch hour for physical activity in working fathers and mothers.[32] One example of a successful approach to increasing physical activity in fathers is the *Healthy Dads, Healthy Kids* intervention. Articles relating to this intervention were excluded from this review as paternal physical activity was explicitly a secondary outcome in these papers,[34–36] which was an exclusion criterion. More research focused specifically on paternal rather than parental physical activity is needed to ensure that both parents can achieve physical activity recommendations.

The majority of parental physical activity research has been conducted with those who have children ≤12 years old. Considering the different challenges that parents may face as their children get older, more primary research is needed around physical activity levels in parents with older children. Following parent–child dyads in existing cohorts as children age could also be of benefit, adding to the limited longitudinal literature in this field. Although some evidence suggests that it is only parents of young children who are significantly less active than non-parents,[8 37] more research is required to confirm whether this is true and if so, to explore why this might be the case.

The proportion of studies using device-based assessment of physical activity has greatly increased from only one study in 2008,[6] to 52 articles in this scoping review. This is a positive development given that adults may change the types of physical activity they engage in when they become parents, shifting from leisure time physical activity to more domestic activities, which may not be adequately captured by self-reported measures.[38] Physical activity in women with young children may be more likely to be sporadic, unstructured and spontaneous, which is better captured using device-based measures rather than self-report.[38] Ideally, physical activity in this population would be measured both by device-based assessment to capture intensity, frequency and duration, and self-report to investigate the domains and types of physical activity.[39] This would allow better characterisation of physical activity in parents and contribute important information relevant to intervention development.

Most of the observational and interventional articles in this scoping review assessed or targeted individual-level correlates and determinants (eg, age of parent, employment status, enjoyment of physical activity and self-efficacy), which mirrors articles relating to adult physical activity in general.[21] No observational articles in this paper investigated the association of parental physical activity with policy, the wider societal level in the SEM, and only two interventional articles related to studies which targeted the wider societal level to increase parental physical activity.[40 41] Thus, more research is needed in this area, especially considering that this has the potential to change the physical activity behaviours of larger numbers of people.[21] Further research is also required into interpersonal (eg, social support and BMI of child) and environmental factors (eg, aesthetics, job flexibility and road safety) in relation to physical activity, as well as into behavioural factors (eg, past physical activity behaviour and smoking), which are the least studied of the individual-level variables.

Considering the large number of observational and interventional articles, a systematic review of these bodies of evidence would be useful to provide researchers and policymakers with information about what factors are associated with the physical activity of parents and what levels of the SEM should be targeted to increase parental physical activity. Bellows-Riecken and Rhodes did conduct a systematic review of studies investigating factors associated with parental physical activity.[6] However, this scoping review shows that there have been many articles published since. Synthesis of the combined evidence could provide answers where previously evidence was inconclusive, for example, in the case of the associations between partner status, employment status or theoretical models and parental physical activity.[6]

In qualitative articles, some areas of research have been addressed frequently, such as barriers and facilitators of parental physical activity. A systematic review of these articles would be beneficial to synthesise the findings and identify consistent barriers and facilitators across the literature base. Primary qualitative research is also needed to advance less frequently explored research questions, such as physical activity changes for mothers and fathers in the transition to parenthood and as their children get older, and experiences of co-participation in physical activity with children. The former topic could help to explain differences in physical

activity levels between parents and non-parents, and between parents of children of different ages.[37] The latter could provide evidence as to whether co-participation would be a suitable target for increasing physical activity levels of parents and children simultaneously.[42]

## Strengths and limitations

This review encompasses a wide range of articles and was conducted rigorously and systematically following a predefined protocol and existing guidance for scoping reviews. It therefore provides detailed information relevant to future parental physical activity research.

Although a systematic search was conducted, it is possible some articles may have been missed as in all reviews. However, this would not affect the major aim of the scoping review which is to provide an overview of the literature base. The review is also limited to English language articles and does not include grey literature. Articles relating to parents in clinical populations themselves or with children in clinical populations were also beyond the scope of this review, but would be of interest to those working in these fields and may be important to collate and examine in future reviews. It is also important to note that the definitions of a parent vary across the literature, but in most cases, parents were mothers or fathers of children, whether biological, adoptive, foster or step, consistent with our definition in the introduction.

## Recommendations for future research

The nature of a scoping review precludes conclusions being drawn as to what factors are and are not associated with parental physical activity, the effectiveness of interventions and recurring themes in the qualitative literature. However, this scoping review has identified sets of articles which could be collated for either a qualitative or quantitative systematic review of parental physical activity. It would be important for researchers conducting such reviews to take into account the quality of studies, which was beyond the scope of this review, and to consider the variety of populations represented in these studies. Several gaps or lesser studied areas in the literature base have also been identified: longitudinal studies with device-measured physical activity, paternal physical activity research, studies conducted in continents other than North America and research focused on specific subgroups of parents, such as single parents, or parents of older children >12 years-old). Behavioural, interpersonal, environmental and wider societal factors also require greater investigation in quantitative research. Topics requiring further exploration in the qualitative literature include co-participation and changes in physical activity since becoming a parent.

## CONCLUSIONS

A wide array of quantitative and qualitative research has been conducted in relation to parental physical activity. However, much of the research relates to mothers, the general population in North America and parents of younger children. There is already scope to conduct systematic reviews of related articles, such as the barriers and facilitators to parental physical activity. However, this review has also highlighted understudied areas which require further primary research, for example, related to paternal physical activity, to fully understand parental physical activity to inform future interventions and policies.

**Acknowledgements** The authors would like to thank Veronica Phillips from the University of Cambridge Medical Library who provided advice on the literature searches.

**Contributors** RFS, EvS, KRH and KE designed the study. RFS performed the literature searches. RFS conducted the title and abstract screening after RFS and KE had assessed 1000 in duplicate. KRH and RFS then assessed titles and abstracts initially found to finalise those articles to be screened at full-text stage. RFS, KRH and EvS conducted full-text screening, and RFS conducted data extraction which was checked by KE or EvS. RFS drafted the manuscript. All authors contributed to the interpretation of the results and critically reviewed the manuscript. All authors read and approved the final manuscript. RFS is the guarantor for this paper.

**Funding** RFS is funded by an ESRC studentship (RG84395). The work of EvS is supported by the Medical Research Council (grant number MC_UU_00006/5). KRH is funded by the Wellcome Trust (grant number 107337/Z/15/Z). The work of KE is funded by NIHR School for Public Health Research (grant number SPHR-PROG-WSBT-CS2).

**Competing interests** None declared.

**Patient and public involvement** Patients and/or the public were not involved in the design, or conduct, or reporting, or dissemination plans of this research.

**Patient consent for publication** Not applicable.

**Provenance and peer review** Not commissioned; externally peer reviewed.

**Data availability statement** All data relevant to the study are included in the article or uploaded as supplementary information. All data extracted from included articles are included in the paper or uploaded as supplementary information.

**ORCID iD**
Rachel F Simpson http://orcid.org/0000-0002-7164-6955

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
