## [Reviewer comments · BMJ Open]

ARTICLE DETAILS

TITLE (PROVISIONAL)	What research evidence exists about physical activity in parents? A systematic scoping review
AUTHORS	Simpson, Rachel; Hesketh, Kathryn; Ellis, Kate; van Sluijs, Esther

VERSION 1 – REVIEW

REVIEWER	Dee Dlugonski University of Kentucky
REVIEW RETURNED	13-Aug-2021

GENERAL COMMENTS	This is a much needed scoping review of the parental physical activity literature that will help to drive future research in areas where there are gaps. I commend the authors for such a large and detailed review. I have provided a few comments that I hope will strengthen the manuscript. Abstract There seem to be problems with the numbers of studies reported. It may be that studies are counting in multiple categories, but this is a bit confusing as a reader. It is unclear whether the 27 studies with multiple study designs are included in the quantitative and qualitative numbers. Cross sectional, longitudinal, and interventional studies do not add to 173. Qualitative and quantitative numbers do not equal the total number of studies included. Please clarify. Introduction The introduction is well written and provides a clear rationale for why this scoping review is needed among parents. Stage 1 Could the authors elaborate on whether the factors of the SEM are being explored as possible predictors of parental physical activity or outcomes/benefits/consequences of parental physical activity? Unique benefits of physical activity for parents were mentioned in the introduction but it is unclear whether these benefits are being explored in the review. Stage 2 In Table 2, for the focus/outcomes section under qualitative, did the purpose of the study have to be related to physical activity or were studies included that asked some questions about physical activity without an explicit purpose on physical activity? Results The numbers in this first paragraph do not seem to match the abstract. Page 18, line 163, please provide summary description of the specific factors that were targeted in the individual, interpersonal, environmental, and societal levels as the authors did for the observational studies. Discussion Page 22, lines 209 – 212. It is unfortunate that the Healthy Dads,
--

	Healthy Kids study was not included in this review given its success for increasing physical activity. Could you provide a rationale for the exclusion of secondary physical activity outcomes in the methods? The inclusion of variables across the SEM was a novel aspect of this review. Please consider expanding on these results in the discussion as well as how they apply to future research.
--	---

REVIEWER	Aidan Searle National Institute for Health Research (NIHR) Biomedical Research Unit in Nutrition, Diet and Lifestyle at the University Hospitals Bristol NHS Foundation Trust and the University of Bristol, Bristol Biomedical Research Centre
REVIEW RETURNED	17-Dec-2021

GENERAL COMMENTS	This manuscript presents a well crafted protocol and scoping review of quantitative and qualitative studies that aims to provide an overview of the current evidence base on parental PA. A range of quantitative and qualitative research has been conducted on parental PA. The review highlights opportunities for evidence synthesis to inform intervention development (such as barriers and facilitators of parental PA) and identifies gaps in the literature, for example, around paternal PA. However, there are some general issues with regard to methodology and evaluating and / or integrating qualitative studies in this context that would further the quality of the submission. What is the the justification for the databases used in the searches - are they comprehensive? Were studies' authors contacted for additional information not included in the published papers or were in possession of non published data of relevance? What is the relative value of quantitative versus qualitative data in this context and how do the approaches speak to each other and inform the outcomes of interest? With regard to theory - are there other competing theories to the SEM that are worthy of consideration in this context? It is stated that qualitative findings would be grouped by questions asked at interview stage - would it me more appropriate to group /evaluate by themes developed in the qualitative analysis? The samples comprising the qualitative studies tend to represent specific populations such as SES / Race, clinical populations, trial evaluations, or clinical groups (i.e diabetes). What are the implications for synthesising /aggregating data from these studies? As far as I can see the quality of individual qualitative studies has not been appraised - how do the authors propose this should be conducted and the dimensions in which the quality of qualitative methods and reporting are measured.
---

REVIEWER	Eva Marie-Louise Denison FHI, Health sevicees
REVIEW RETURNED	21-Dec-2021

GENERAL COMMENTS	My input to the review of this manuscript mainly covers
---

	methodological aspects of the work. This systematic scoping review is conducted "by the book". The authors adhere to the main literature and PRISMA reporting guidelines for scoping reviews. The protocol for the review was registered with the Open Science Framework. The research question on parental participation in physical activity is no doubt relevant to the field of research in physical activity, The authors have summarized a vast literature in a clear way. They have identified knowledge gaps within this literature and indicated possibilities for systematic reviews of effects of interventions and quantitative evidence syntheses of e.g. barriers to physical activity for parents. The authors have discussed strengts and potential limitations of their work. One important potential limitation is the decision to omit checking reference lists and citations of included articles. Given the large number of included publications from searches in four main databases there is only a small probability that the conclusions would be affected in a serious way. I acknowledge that other reviewers with more expertise in the reseach field of physical activity may have comments regarding the subject matter in this scoping review.
--	---

VERSION 1 – AUTHOR RESPONSE

Reviewer 1

Comment	Response	Relevant page number of main manuscript or file
This is a much needed scoping review of the parental physical activity literature that will help drive future research in areas where there are gaps. I commend the authors for such a large and detailed review. I have provided a few comments that I hope will strengthen the manuscript.	Thank you very much.	n/a
Abstract - There seem to be problems with the numbers of studies reported. It may be that studies are counting in multiple categories, but this is a bit confusing as a reader. It is unclear whether the 27 studies with multiple study designs are included in the quantitative and qualitative numbers. Cross-sectional, longitudinal and interventional studies do not add to 173. Qualitative and quantitative numbers do not equal the total	We apologise for the confusion. 27 refers to the number of papers which included more than one of the following study designs: qualitative, longitudinal, cross-sectional, retrospective, interventional. We have now adjusted the abstract to clarify: "27 papers reported on more than one study design".	2

number of studies included. Please clarify.		
Introduction – the introduction is well written and provides a clear rationale for why this scoping review is needed among parents.	Thank you	n/a
Stage 1 – Could the authors elaborate on whether the factors of the SEM are being explored as possible predictors of parental physical activity or outcomes/benefits/consequences of parental physical activity? Unique benefits of physical activity for parents were mentioned in the introduction but it is unclear whether these benefits are being explored in the review.	The reviewer is correct that we do mention the benefits of physical activity for parents in the introduction, but SEM factors are being explored as factors which may influence physical activity levels of parents rather than as outcomes themselves. On page 5, we mention that the SEM is a useful framework to examine factors that influence parental physical activity. We have now clarified this in the aims at the end of the introduction: “The overall aim of this scoping review is therefore to determine the extent and nature of the literature exploring physical activity levels in parents to inform future intervention development. This includes study designs used, ages of children, representation of fathers, assessment method of physical activity, factors in quantitative studies which could influence parental physical activity, according to the SEM, and questions addressed in the qualitative literature.”	6
Stage 2 – In Table 2, for the focus/ outcome section under qualitative, did the purpose of the study have to be related to physical activity or were studies included that asked some questions about physical activity without an explicit purpose on physical activity?	We included qualitative studies where physical activity was a main focus of the paper. We operationalised this as meaning that the purpose of the study had to be related to physical activity or a significant proportion of the paper had to relate to physical activity. In the large majority of cases, the purpose of the study was related to physical activity. We have now clarified this in Table 2 under the focus/ outcome section, “The purpose of the study had to be related to physical activity or a significant proportion of the paper had to focus on physical activity.”	Table 2, p10 and Supplementary File 4

Results – the numbers in the first paragraph do not seem to match the abstract.	We have checked this and the numbers in both the abstract and the first paragraph of Results are both correct. However, the numbers do not match due to different groupings of papers – the abstract numbers related to the number of papers with each study design (including those with multiple study designs), whilst the Results section gave the numbers of papers which had only one study design (as well as giving numbers separately for those which have multiple study designs). We have now adjusted the Results section to match the abstract to avoid confusion: “Of the 213 articles included (Figure 1), 27 reported on more than one study design. 173 (81%) articles reported on quantitative study designs (81 cross-sectional, 26 longitudinal, 76 interventional) and 58 (27%) on qualitative data.”	13
Results – Page 18, line 163, please provide a summary description of the specific factors that were targeted in the individual, interpersonal, environmental, and societal levels as the authors did for the observational studies.	We have now added in this information: “Most interventions (88% of 73 articles) targeted individual factors (e.g. self-efficacy and knowledge of physical activity), followed by interpersonal (49%) (e.g. social support provision), environmental (14%) (e.g. changes to the physical or social environment, such as modifications to the home or work environment), and then wider societal (3%) (e.g. public policy) (see Supplementary File 9 for details by study).”	18
Discussion – Page 22, lines 209-212. It is unfortunate that the Healthy Dads, Healthy Kids study was not included in this review given its success for increasing physical activity. Could you provide a rationale for the exclusion of secondary physical activity outcomes in the Methods.	We have now included this exclusion criterion in the main inclusion and exclusion table (Table 2), rather than just in the supplementary table which provides a more detailed description of criteria applied. We have also included a rationale for exclusion of secondary physical activity outcomes in Table 2 under exclusion criteria related to focus/ outcomes: “Articles in which parental physical activity is explicitly a secondary outcome. These articles were not deemed to have a main focus on parental physical activity.”	Table 2, p10
Discussion – the inclusion of variables across the SEM was a	We are very pleased that you think that this is an interesting part of the review. We have	n/a

novel aspect of this review. Please consider expanding on these results in the discussion as well as how they apply to future research.	considered expanding on these results in the discussion, but since 2 paragraphs of the discussion already relate to this, we would prefer not expand on this section further and have not made any changes to the paper related to this comment.	
---	--	--

Reviewer 2

Comment	Response	Relevant page number of main manuscript or file
The manuscript presents a well-crafted protocol and scoping review of quantitative studies that aims to provide an overview of the current evidence base on parental PA. A range of quantitative and qualitative research has been conducted on parental PA. This review highlights opportunities for evidence synthesis to inform intervention development (such as barriers and facilitators of parental PA) and identifies gaps in the literature, for example, around paternal PA. However, there are some general issues with regard to methodology and evaluating and/or integrating qualitative studies in this context that would further the quality of the submission.	Thank you very much.	n/a
What is the justification for the databases used in the searches – are they comprehensive?	The databases were chosen to provide access to a range of interdisciplinary articles, as mentioned in the Methods on page 9. We have now added in that the range was comprehensive.	9
Were studies' authors contacted for additional information not included in the published papers or were in possession of non-published data of relevance?	We were interested in describing what types of articles had been published, but not in analysing the data ourselves, so we did not contact studies' authors. We have not made any changes to the paper related to this comment.	n/a
What is the relative value of quantitative versus qualitative data in this context and how do the	We have added an explanation for this into the Introduction of the paper:	6

approaches speak to each other and inform the outcomes of interest?	“Qualitative and quantitative research are complementary to each other. They answer different questions about physical activity, for example quantitative research can tell us what factors are associated with parental physical activity, whilst qualitative research can tell us why this is the case.”	
With regard to theory, are there competing theories to the SEM that are worthy of consideration in this context?	There are indeed other theories, for example Social Cognitive Theory or Family Systems Theory. We opted to focus on SEM as this is commonly used and provides a clear framework for the presentation of results. We have not made any changes to the paper related to this comment.	n/a
It is stated that qualitative findings would be grouped by questions asked at the interview stage - would it be more appropriate to group/ evaluate by themes developed in qualitative analyses?	Findings were grouped by questions as this fits with the aims of this scoping review. Identification of recurring themes would be appropriate in a qualitative systematic review as part of future research. We have included a statement about this in the discussion: “The nature of a scoping review precludes conclusions being drawn as to...recurring themes in the qualitative literature.” We have not made any further additions to the paper.	25
The samples comprising the qualitative studies tend to represent specific populations, such as SES/ race, clinical populations, trial evaluations, or clinical groups (i.e. diabetes). What are the implications for synthesising/ aggregating data from these studies?	We think that this should be a consideration for a future systematic review rather than the current scoping review. In a scoping review, the focus is on what research has been conducted, rather than synthesising the data to find themes. We have added a statement into the discussion to highlight the importance of considering specific populations: “It would be important for researchers conducting such reviews to take into account the quality of studies, which was beyond the scope of this review, and to consider the variety of populations represented in these studies.”	25
As far as I can see, the quality of individual studies has not been appraised – how do the authors propose this should be conducted and the dimensions in which the quality of qualitative methods and reporting are measured?	As mentioned in the Methods, p11, the assessment of qualitative studies is beyond the scope of this systematic scoping review. We have now added in a reference to Arksey and O’Malley 2005 that quality assessment in scoping reviews is optional. In the Discussion, we have also noted the importance of assessing the quality of studies for those conducting a systematic	11, 25

	review: “It would be important for researchers conducting such reviews to take into account the quality of studies, which was beyond the scope of this review, and to consider the variety of populations represented in these studies.”	
--	--	--

Reviewer 3

Comment	Response	Relevant page number of main manuscript or file
My input to the review of this manuscript mainly covers methodological aspects of the work. This systematic scoping review is conducted “by the book”. The authors adhere to the main literature and PRISMA reporting guidelines for scoping reviews. The protocol for the review was registered with the Open Science Framework. The research question on parental participation in physical activity is no doubt relevant to the field of research in physical activity. The authors have summarised a vast literature in a clear way. They have identified knowledge gaps within this literature and indicated possibilities for systematic reviews of effects of interventions and quantitative evidence synthesis of e.g barriers to physical activity for parents.	Thank you	n/a
The authors have discussed strengths and potential limitations of their work. One important limitation is the decision to omit checking reference lists and citations of included articles. Given the large number of publications from searches in four main databases, there is only a small probability that the conclusions would be affected in a serious way. I acknowledge that other reviewers with more expertise in the research area of physical activity may have comments regarding the subject matter in this scoping review.	We agree that given the large number of publications from four main databases, there is little chance that conclusions would be affected by the inclusion of reference list and citation checking for other articles to be included.	n/a

VERSION 2 – REVIEW

REVIEWER	Dee Dlugonski University of Kentucky
REVIEW RETURNED	03-Mar-2022

GENERAL COMMENTS	Nice work on the revised manuscript. I have no further comments.
--

REVIEWER	Aidan Searle National Institute for Health Research (NIHR) Biomedical Research Unit in Nutrition, Diet and Lifestyle at the University Hospitals Bristol NHS Foundation Trust and the University of Bristol, Bristol Biomedical Research Centre
REVIEW RETURNED	15-Feb-2022

GENERAL COMMENTS	Having read all the reviewer comments I am satisfied that the authors have satisfactorily revised the manuscript and is now acceptable for publication.
---